# Optimizing Osimertinib for NSCLC: Targeting Resistance and Exploring Combination Therapeutics

**DOI:** 10.3390/cancers17030459

**Published:** 2025-01-29

**Authors:** Yan-You Liao, Chia-Luen Tsai, Hsiang-Po Huang

**Affiliations:** 1Department of Medicine, National Taiwan University College of Medicine, Taipei 100233, Taiwan; b10401057@ntu.edu.tw; 2Graduate Institute of Medical Genomics and Proteomics, National Taiwan University College of Medicine, Taipei 100233, Taiwan; d11455001@ntu.edu.tw

**Keywords:** non-small-cell lung cancer, osimertinib resistance, combination therapy

## Abstract

Osimertinib has revolutionized EGFR-mutant NSCLC therapy by specifically targeting the T790M mutation, yet acquired resistance remains inevitable. Mechanisms include tertiary EGFR mutations such as C797S, bypass pathway activation (MET, HER2, HER3), and histologic transformations. Emerging combination strategies that integrate osimertinib with chemotherapy, angiogenesis inhibitors, MET-targeting agents, immunotherapies, and other molecularly tailored drugs show the potential to circumvent resistance and prolong clinical benefits. Real-time molecular monitoring, including liquid biopsy and single-cell analyses, is crucial for the early detection of resistant clones and guiding therapeutic adjustments. While ongoing clinical trials will clarify optimal combinations and sequencing, a personalized approach that incorporates biomarker-driven patient selection and adaptive treatment paradigms remains essential. By harnessing diverse strategies against parallel signaling pathways and immune evasion, these regimens aim to extend survival and preserve quality of life. Continued collaboration among researchers, clinicians, and patients will accelerate the translation of these advances into practice.

## 1. Background

Lung cancer is the second-most common cancer worldwide and the leading cause of death among all cancer types [1,2], with non-small-cell lung cancer (NSCLC) accounting for 80–90% of cases. Around 60% of patients are diagnosed with advanced or metastatic disease, historically treated with platinum-based chemotherapy as the primary standard [3,4]. However, understanding the molecular underpinnings of NSCLC led to the development of epidermal growth factor receptor (EGFR) tyrosine kinase inhibitors (TKIs), dramatically improving treatment outcomes compared to chemotherapy [4,5,6,7,8,9]. First- and second-generation EGFR TKIs (e.g., gefitinib, erlotinib, afatinib, dacomitinib) offered significant benefits, but were hindered by acquired resistance—most commonly the T790M mutation—within about one year [10,11,12,13,14,15,16,17]. Third-generation TKIs such as osimertinib were designed to overcome T790M-mediated resistance, irreversibly binding to mutant EGFR and blocking downstream signaling [18,19,20,21,22,23,24,25,26,27].

Clinical trials have confirmed the central role of osimertinib in treating EGFR-mutant NSCLC. In the AURA studies, it prolonged progression-free survival (PFS) and achieved high response rates in patients with T790M-positive disease [19,28,29,30]. AURA3 compared osimertinib to platinum–pemetrexed chemotherapy, demonstrating superior PFS (10.1 vs. 4.4 months) and overall response rate (ORR) (71% vs. 31%), with strong activity against brain metastases [31,32,33,34,35,36,37]. Although early overall survival (OS) data were similar, subsequent analyses that accounted for high crossover favored osimertinib [34,35,37]. The FLAURA trial established osimertinib as a first-line therapy for EGFR-mutant advanced NSCLC [24,38,39,40,41,42,43], revealing significant improvements in PFS (18.9 vs. 10.2 months) and OS (38.6 vs. 31.8 months) compared to first-generation TKIs (gefitinib/erlotinib) [24]. Real-world data in 56 patients supported these findings, though 55% were ineligible under FLAURA criteria [44]. The median time to treatment discontinuation was longer for FLAURA-eligible patients (31.1 vs. 12.2 months) [44]. Re-biopsy findings indicated on- and off-target resistance mechanisms, emphasizing the need for diverse post-progression strategies [44].

In the adjuvant setting, the LAURA trial demonstrated significantly longer significantly longer PFS than placebo in patients with unresectable stage III EGFR-mutant NSCLC [45]. The percentage of patients who were alive and progression-free at 12 months was 74% vs. 22% with placebo [45]. On the other hand, the ADAURA trial evaluated osimertinib in resected stage IB–IIIA EGFR-mutant NSCLC, demonstrating a substantial disease-free survival advantage (22.1 vs. 14.9 months) [46]. The final results of the ADAURA trial confirmed a five-year OS rate of 85% for stage II–IIIA patients on osimertinib versus 73% on placebo [47]. Importantly, a severe COVID-19-related pneumonia occurred post-data cutoff, but was deemed unrelated to osimertinib [46]. The patient fully recovered. In a retrospective study conducted during the COVID-19 pandemic, the delayed diagnosis caused by COVID-19 was identified as a key factor leading to the suboptimal PFS (18.0 months) of osimertinib treatment [48]. In contrast, another study, leveraging its inhibitory effect on EGFR, proposed osimertinib as a potential therapeutic agent for COVID-19 [49]. To recap, these findings position osimertinib as a cornerstone in EGFR-mutant NSCLC management—first-line, T790M-positive, and adjuvant. Ongoing research into emerging resistance mechanisms and broader, more inclusive clinical trials will be crucial to refine and extend the benefits of osimertinib across diverse patient populations [50,51,52,53,54].

## 2. The Mechanisms of NSCLC Osimertinib Resistance

Despite these promising outcomes, resistance to osimertinib remains a critical issue, presenting new challenges for managing NSCLC [27,55,56,57,58]. Although osimertinib prolongs PFS and has robust intracranial activity that greatly benefits CNS-metastatic patients [42,47,59,60,61,62], its cost-effectiveness in first- or second-line therapy is a concern in many countries [58,63,64,65]. Furthermore, adverse effects such as hyponatremia, as well as severe side effects associated with structurally similar drugs like furmonertinib and rezivertinib, complicate its clinical use [66,67,68]. Although osimertinib offers durable disease control for many, progression inevitably occurs in most patients, making it essential to understand the resistance mechanisms that emerge [69,70,71].

## 3. On-Target (EGFR-Dependent) Mechanisms and Corresponding Therapeutic Strategies

### 3.1. Tertiary Mutations at C797

Resistance to third-generation EGFR TKI, including osimertinib, often arises from tertiary mutations in the EGFR gene. One of the most prevalent mutation sites is C797 (Table 1), located at the ATP-binding site and encoded by exon 20 [57,72,73,74,75]. Mutations at this site, such as the predominant C797S [76,77,78] or the rarer C797G [79], disrupt osimertinib’s mechanism of covalent binding to EGFR. Consequently, tumors harboring C797S frequently show reduced sensitivity to osimertinib, though first- or second-generation TKIs may retain efficacy if C797S and T790M are present in a trans (rather than cis) configuration [74,76,80,81,82,83,84]. Moreover, studies have shown that the C797S mutation can also induce resistance to other third-generation EGFR TKI, such as olmutinib, lazertinib, and abivertinib [72,85,86,87].

Circulating tumor DNA (ctDNA) analyses have been pivotal in delineating resistance mechanisms [77,88,89,90,91,92,93,94,95]. In the AURA3 trial, 49% of patients lost the T790M mutation at progression, indicating a shift toward alternative pathways [35,36,37]. Among T790M-positive individuals on osimertinib, 63% no longer had detectable T790M upon progression, commonly coinciding with histologic transformations, KRAS mutations, or gene fusions [65]. T790M “loss” often heralded shorter treatment durations (6.1 months vs. 15.2 months) and poorer survival outcomes, with the shorter PFS having a median of 2.6 months [22,23,65]. Importantly, T790M is generally absent when osimertinib is used as first-line therapy, hinting that T790M-driven resistance may become less relevant as osimertinib is deployed earlier [74]. However, in second-line settings, C797S remains a predominant EGFR-dependent resistance mechanism. Clinical trials like AURA3 and FLAURA reported C797 mutations in 15% and 7% of cases, respectively [32,40], while real-world studies cited rates between 11% and 29% [50,78,96,97,98].

### 3.2. Other EGFR Mutations

While the C797S mutation is a major focus, other tertiary EGFR mutations also play a role in resistance (Table 1). G796 mutations (G796R, G796S, G796H, G796D), similarly to those at C797, disrupt the binding of TKIs to the EGFR kinase domain [39,98,99,100,101,102]. Additionally, the L792 residue, located near C797 and T790, is frequently mutated (L792H, L792Y, and L792F) in osimertinib-resistant cases, with L792H being particularly common and a potent driver of resistance [99]. L792 mutations tend to occur in cis with T790M, but in trans with C797S, offering important insights for clinicians who are assessing T790M- and C797S-related resistance mechanisms [100,103]. Though many of these findings are based on computational models and in vitro studies [104,105,106], further investigation is needed to confirm these effects. In addition to C797 and L792 mutations, several rare EGFR mutations—including L718, G724, and G796—have been implicated in osimertinib resistance [106,107,108,109]. The L718 residue, positioned in a peripheral region of the ATP-binding site encoded by exon 18, undergoes mutations (such as L718Q and L718V, while L718Q mutation remains sensitive to quinazoline-based inhibitors, such as afatinib and gefitinib) that impedes its interaction with the phenyl ring of osimertinib, contributing to resistance [106,107]. Patients with EGFR mutations that combine L858R, T790M, and L718 are typically resistant to first- and second-generation TKIs, whereas those with combined L858R and L718 mutations often respond to afatinib [39,106,107,110].

Similarly, the G724S mutation, commonly observed in second-line osimertinib-treated cases, alters the glycine-rich loop in EGFR, hindering covalent bonding with osimertinib [108,111]. In vitro studies have shown that cells with the G724S mutation respond to afatinib, especially when T790M is absent [39,111,112]. One study further noted that the G724S mutation induces resistance in cases with exon 19 deletion mutations, while cancers with L858R/G724S mutations remain sensitive to osimertinib [112]. Additional rare mutations, including G719, S768, and other G796 variants, have also been associated with osimertinib resistance [39,59,113,114]. These mutations reduce osimertinib-binding efficacy. Lastly, dual T790M–M766Q mutations render cells resistant to osimertinib, although these cells retain sensitivity to HER2 and EGFR dual inhibitors, such as neratinib and poziotinib [115].

**Table 1 cancers-17-00459-t001:** On-target mechanisms of osimertinib resistance and therapeutic strategies in EGFR-mutant NSCLC.

Mechanism Category	Approximate Prevalence	Therapeutic Strategies	References
**Tertiary mutations at C797** (e.g., C797S, C797G)	11–29% (real-world); ~15% (AURA3); ~7% (FLAURA)	First/second-generation TKIs if T790M and C797S are trans-mutated Fourth-generation EGFR TKIs, combination therapies under investigation	[32,40,50,57,72,73,74,75,76,77,78,79,80,81,82,83,84,85,86,87,96,97,98]
**Other EGFR mutations** L792 (L792H/Y/F), L718 (L718Q/V), G724S, G796 (G796R/S/H/D)	Rare (varies by cohort, typically <5%)	Switching to afatinib for certain L718Q or G724S cases Combination EGFR-targeted therapies (e.g., dual EGFR–HER2 inhibitors)	[39,59,98,99,100,101,102,103,104,105,106,107,108,109,110,111,112,113,114,115]
**Exon 20 insertions** (EGFR20ins)	10–12% of EGFR-driven NSCLC (all lines)	Limited sensitivity with early-generation TKIs Novel EGFR inhibitors under development (e.g., mobocertinib, amivantamab)	[116,117,118,119,120]

### 3.3. Exon 20 Insertions

Beyond these well-characterized EGFR mutations, recent attention has turned to *EGFR* exon 20 insertion (EGFR20ins)-positive NSCLC, which accounts for 10% to 12% of EGFR-addicted tumors [116,117,118]. Historically, EGFR20ins variants have shown limited sensitivity to early-generation EGFR TKIs, in part because they share an ATP-binding pocket resembling wild-type *EGFR* [119,120]. Meanwhile, evidence suggests that uncommon *EGFR* exon 19 deletions such as L747_A750>P can also exhibit reduced sensitivity to osimertinib compared to the common E746_A750del, resulting in shorter PFS for patients harboring this variant [121].

## 4. Off-Target (EGFR-Independent) Resistance Mechanisms and Corresponding Therapeutic Strategies

In addition to mutations that reduce affinity for EGFR inhibitors, tumor cells can develop resistance through mechanisms (Table 2) such as histologic transformation or activating other receptor tyrosine kinases (RTKs) and alternative downstream pathways (Figure 1).

### 4.1. MET Amplification

*MET* amplification is a well-established mechanism of resistance to EGFR TKI, including third-generation agents such as osimertinib, accounting for 3% to 19% of resistance cases [32,40,96,122,123,124,125,126,127]. By activating ErbB3 phosphorylation independently of EGFR, *MET* amplification sustains downstream PI3K/AKT signaling and cell survival in the presence of EGFR inhibitors [128,129]. Additional findings indicate that high levels of death receptor 4 in *MET*-amplified cells impede apoptosis, contributing to osimertinib resistance [130]. A *GOPC-ROS1* rearrangement has also been linked to MET-driven resistance, but remains sensitive to the MET inhibitor crizotinib [131].

*MET* encodes the c-Met receptor tyrosine kinase, which binds hepatocyte growth factor (HGF) secreted by mesenchymal cells. Ligand binding triggers PI3K, JAK/STAT, and MAPK signaling pathways, involving miR-21 and driving processes like cell morphogenesis and mitosis [132,133,134,135]. Through amplification, MET can bypass EGFR blockade, directly activating downstream effectors [133,135,136]. Importantly, *MET* amplification arises as either focal amplification or chromosome 7 polysomy [137,138]. Focal amplification, often propelled by breakage–fusion–bridge cycles, typically carries oncogenic potential, whereas polysomy, involving multiple copies of chromosome 7, may not drive tumor progression [139,140,141,142,143]. Distinguishing these events requires a *MET* gene copy number threshold (≥5) or a *MET*-to-*CEP7* ratio ≥ 2, as determined by fluorescence in situ hybridization (FISH) [139,140,141,142,143].

Although next-generation sequencing (NGS) can detect *MET* amplification, it does not always quantify chromosome 7 copy numbers. Consequently, FISH remains the preferred method when NGS data are ambiguous [142,144]. *MET* amplification is present in 5–24% of progression events after osimertinib, appearing more frequently when osimertinib is used first-line. By contrast, C797S mutations predominate when osimertinib is given second-line [72,74,145]. In second-line settings, *MET* amplification often coexists with other genomic alterations (e.g., T790M loss, C797S, or amplifications in *CDK6* and *BRAF*) [104,146]. Preclinical studies indicate that combining a MET inhibitor with afatinib can overcome resistance in *MET*-amplified EGFR-mutant cell lines [147].

It is also essential to distinguish *MET* amplification from *MET* exon 14 (METex14)-skipping mutations, which occur in 3% to 4% of lung adenocarcinomas. These mutations reduce MET protein turnover and can induce resistance to first-generation EGFR TKIs like erlotinib, but are not frequently implicated in osimertinib resistance [148,149]. HGF overexpression, associated with MET, further promotes resistance by stimulating MET activity independently of amplification [150]. Cancer-associated fibroblasts (CAFs) in osimertinib-resistant models secrete elevated levels of HGF, potentially driving tumor survival [135,136]. TGF-β–regulated CAFs with high HGF levels appear critical to the osimertinib response, though further studies are needed to fully clarify HGF’s role in third-generation EGFR-TKI resistance [136].

### 4.2. HER2 Amplification and HER3 Upregulation

*HER2* and *HER3* have emerged as crucial drivers of resistance in EGFR-mutant NSCLC, particularly in the setting of third-generation EGFR TKIs such as osimertinib. Although *HER2* amplification occurs in roughly 1% of NSCLC cases, its incidence increases post-osimertinib: data from trials like AURA and FLAURA confirm *HER2* amplification rates of about 5% in second-line and 2% in first-line resistance, often in the absence of T790M mutations [32,40]. *HER2* point mutations—including exon 20 insertions—and the HER2D16 splice variant further contribute to resistance by altering the kinase domain or bypassing EGFR inhibition [119,120,151]. While combining pan-HER inhibitors with osimertinib can counter some *HER2*-driven resistance, clinical efficacy remains limited and necessitates further exploration [151,152].

Attempts to overcome *HER2*-mediated resistance via targeted therapy have yielded mixed outcomes. The TRAEMOS trial, which combined osimertinib with trastuzumab emtansine (T-DM1) in patients showing *HER2* overexpression post-osimertinib, revealed manageable adverse events such as fatigue, diarrhea, and nausea, but only a 4% ORR and a median PFS of 2.8 months [153]. These modest results suggest that while EGFR/HER2 co-targeting holds theoretical appeal, additional research is required to refine these strategies [152]. HER3, although lacking intrinsic kinase activity, plays a pivotal role in resistance by dimerizing with other ErbB receptors—particularly HER2—to robustly activate the PI3K/AKT pathway [154,155,156]. Upregulated HER3 can be found in osimertinib-resistant cells at levels several times higher than in sensitive cells, reinforcing its importance as a resistance mediator [157,158]. Intriguingly, osimertinib itself may induce IRE1α-dependent HER3 upregulation, driving both resistance and immune activation via macrophage recruitment and cGAS-STING signaling [159].

Against this backdrop, novel HER3-targeting strategies show promise. Patritumab deruxtecan (HER3-DXd), an antibody–drug conjugate linking a HER3-directed monoclonal antibody to a topoisomerase I inhibitor, exerts dual actions: inhibiting the PI3K/AKT pathway while delivering a cytotoxic payload [128,160,161,162,163]. In the phase II HERTHENA-Lung01 trial, HER3-DXd produced an ORR of 29.8% and a median PFS of 5.5 months in patients who had progressed on EGFR TKIs and platinum-based chemotherapy, regardless of *HER3* expression levels or specific resistance mechanisms [164,165]. Notably, it achieved a 33.3% ORR in non-irradiated brain metastases, addressing a critical clinical need [165]. These encouraging data supported the launch of the phase III HERTHENA-Lung02 trial comparing HER3-DXd to platinum-based chemotherapy in patients resistant to third-generation EGFR TKIs [166]. Moreover, early-phase findings have demonstrated both an acceptable safety profile and substantial antitumor activity, positioning HER3-DXd as a promising candidate for patients who have exhausted conventional therapies [164,165,166].

### 4.3. Abnormalities in Cell Proliferation and Apoptosis-Related Factors

Resistance to osimertinib in EGFR-mutant NSCLC arises from diverse processes regulating cell proliferation, apoptosis, and cell cycle control. One significant driver is *FGFR1* amplification, which enhances tumor growth and complicates therapy [167,168]. Dual targeting of FGFR and AKT has shown promise in *FGFR1*-overexpressing cells, although challenges persist under hypoxic conditions and *SHISA3* inactivation, both of which further upregulate *FGFR1* [167,168,169,170]. Another important mechanism involves IGF1R activation driven by insulin-like growth factor 2 and forkhead box A1 overexpression [171,172]. In addition, non-coding RNAs, such as hsa_circ_0005576 and hsa_circ_PPAPDC1A, reinforce IGF1R signaling in NSCLC cells via miR-512-5p/IGF1R and miR-30a-3p/IGF1R, respectively [173,174]. These findings provide a rationale for evaluating IGF1R inhibitors alongside osimertinib in clinical settings.

AXL, a receptor belonging to the TAM (TYRO3, AXL, and MERTK) family, is another key contributor to drug resistance in EGFR-mutant NSCLC [175,176,177]. Analyses have correlated high AXL expression with osimertinib resistance, influenced by epigenetic factors and p53 status [15,178,179]. Preclinical evidence shows that inhibiting AXL (e.g., using anlotinib) in combination with osimertinib can delay or counteract resistance [175,180,181]. Further approaches involve antibody–drug conjugates targeting AXL or dual AXL–MET inhibitors, both of which have displayed significant efficacy in preclinical models and are moving toward clinical trials [182,183]. A multifaceted therapy that blocks multiple pathways—such as osimertinib plus an AXL inhibitor (ONO-7475) and an FGFR inhibitor (BGJ398)—has demonstrated marked antitumor effects in high-AXL-expressing, EGFR-mutant NSCLC cells [168,170,184]. This triple combination significantly amplified apoptosis by boosting BIM levels and reduced cell viability compared to dual therapy. In xenograft models, triple inhibition strongly suppressed tumor regrowth, suggesting that initial blockade of FGFR1 may be pivotal for preventing resistance [170].

Apoptotic regulation via the BCL-2 family is also essential for EGFR-TKI success [185]. BIM, a pro-apoptotic protein, is a crucial mediator of cell death: a deletion polymorphism affecting *BIM*—common in about 21% of East Asians—has been linked to worse PFS and OS in patients receiving osimertinib [186]. Histone deacetylase inhibitors (e.g., vorinostat and panobinostat) may restore BIM function and help overcome polymorphism-related resistance [187,188]. Another strategy involves navitoclax, a BCL-2/BCL-xL inhibitor tested in combination with osimertinib in a phase IB trial (ETCTN 9903) [189]. This combination exhibited a 100% objective response rate and a median PFS of 16.8 months, with manageable side effects such as lymphopenia, fatigue, and thrombocytopenia [189]. Early thrombocytopenia confirmed navitoclax’s target engagement and highlighted BCL-2/BCL-xL blockade as a viable means of enhancing osimertinib’s pro-apoptotic effects [189].

Finally, cell cycle dysregulation plays a prominent role in osimertinib resistance. Amplifications in genes encoding cyclins D1, D2, E1, and CDK4/6, along with *CDKN2A* deletions, frequently occur in relapsing patients and correlate with shorter PFS [190]. Phosphorylation of retinoblastoma protein (Rb) via CDK4/6 is critical for G1-to-S phase progression, thereby sustaining proliferation in resistant cells [191,192,193]. Polo-like kinase 1 (PLK1)—a regulator of G2/M transition—has emerged as another potential target, with PLK1 inhibitors demonstrating synergy with osimertinib [194]. Preclinical models show that combining osimertinib with CDK4/6 inhibitors (abemaciclib or palbociclib) impedes Rb phosphorylation, arrests the cell cycle in G1, and curbs tumor growth [195,196]. A phase II study (NCT04545710) is underway to further assess the safety and effectiveness of this approach.

### 4.4. Abnormal Activation of Downstream Proliferative Signaling Pathways

Mutations in genes within downstream signaling pathways, particularly those involving PI3K/AKT and RAS/RAF/MEK/ERK, are pivotal in driving EGFR-TKI resistance independently of EGFR itself [197,198]. PIK3CA, KRAS, and BRAF alterations frequently emerge in osimertinib-resistant NSCLC, making them important targets for precision therapy. Liquid biopsy has proven valuable for monitoring these resistance mutations in real time, reflecting the heterogeneous genomic landscape across multiple tumor sites [199]. Within the PI3K/AKT/mTOR axis, *PIK3CA* mutations such as E454K and E542K enhance kinase activity, bolstering tumor cell proliferation and survival [200]. Loss of the tumor suppressor PTEN, which negatively regulates PI3K/AKT signaling, also contributes to resistance: various studies link PTEN downregulation to enhanced pathway activation [201,202]. Mechanistically, PTEN loss can be triggered by neurofibromin 2b-mediated proteasomal degradation via GSK-3β, underscoring the complexity of this regulatory network [203]. Blocking processes that degrade PTEN or pharmacologically restoring its function could reinforce EGFR-TKI sensitivity [92,202].

Parallel resistance mechanisms involve the RAS/RAF/MEK/ERK (MAPK) pathway, which governs cell growth and differentiation [168,204,205,206]. Emerging data suggest that the downregulation of hsa_circ_0004339 in NSCLC promotes MAPK pathway activation, fostering aggressive tumor behaviors [207]. Clinically, mutations and amplifications in *NRAS* and *KRAS* have been detected in preclinical EGFR-mutant NSCLC models treated with osimertinib [104,208], and the FLAURA and AURA3 trials identified *NRAS* and *KRAS* mutations in up to 3% and 1% of progression cases, respectively [40,50,96,209]. Novel agents targeting KRAS G12C, including sotorasib and adagrasib, have shown promise in advanced NSCLC. Sotorasib has already been approved by the Food and Drug Administration (FDA) for this indication [210,211]. Additionally, BRAF mutations—most notably V600E—drive resistance and tumor progression in some patients [212]. Targeted strategies using dabrafenib (a BRAF V600E inhibitor) and trametinib (an MEK inhibitor) combined with osimertinib have yielded encouraging outcomes, with one study reporting an objective response rate of 61.5% and a median PFS of 13.5 months in patients harboring dual EGFR and BRAF V600E alterations [213]. Other BRAF mutations, such as G469A, may regain sensitivity to osimertinib upon MEK inhibition [214]. In vitro and in vivo data also support combining osimertinib with BRAF inhibitors (e.g., encorafenib, vemurafenib) or trametinib to circumvent resistance [215,216,217,218].

MEK, a vital MAPK component, has drawn attention through inhibitors like trametinib, selumetinib, and binimetinib—some of which have received FDA approval [208,219,220]. Resistance to osimertinib can arise from MEK/ERK-driven reductions in BIM or increases in MCL-1, highlighting the benefit of the dual blockade [221,222]. Indeed, combining osimertinib with trametinib has shown efficacy in overcoming KRAS G12V-mediated resistance, including in leptomeningeal metastasis models [113,223,224]. Nanoformulations co-delivering osimertinib and selumetinib have also demonstrated the capacity to surmount acquired resistance [225,226]. Nonetheless, secondary pathway activation (e.g., Hedgehog) and variable baseline ERK phosphorylation may limit the durability of MEK-targeted strategies [206,227,228]. The TATTON study affirmed the feasibility of osimertinib paired with selumetinib, testing different dosing schedules to optimize safety and efficacy [229].

### 4.5. Histologic Transformation and Epithelial-to-Mesenchymal Transition (EMT)

Histologic transformation is increasingly recognized as an underappreciated mechanism of resistance to osimertinib in EGFR-mutant NSCLC [230,231,232,233,234]. While liquid biopsies facilitate diagnosis and disease monitoring, they often fail to capture these pivotal shifts in tumor histology [230]. Numerous case reports document transformations from adenocarcinoma to small-cell lung cancer (SCLC), squamous cell carcinoma, large-cell neuroendocrine carcinoma, or sarcomatoid carcinoma [85,91,235,236]. Approximately 2% to 15% of patients on first-line osimertinib exhibit such changes, retaining the original EGFR-sensitizing mutations [237]. One study noted that 15% of patients receiving first-line osimertinib and 14% of those on later-line regimens underwent histologic transformation, underscoring the necessity of tissue biopsy for accurate assessment [122]. However, frequent biopsies are invasive and impractical given tumor heterogeneity, highlighting an urgent need for molecular markers to distinguish transformation events [78,237]. Research implicates the dysregulation of PI3K/AKT/mTOR and NOTCH pathways in driving shifts from adenocarcinoma to SCLC or squamous cell carcinoma, making them potential therapeutic targets [231,238].

Genetic profiling reveals that SCLC transformation commonly involves *RB1* and *TP53* loss [239,240]. Patients harboring baseline *EGFR*, *RB1*, and *TP53* mutations, which are present in about 5% of EGFR-mutant NSCLC, face a higher risk of transformation and poorer outcomes [91,234,239,240]. The presence of *RB1* and *TP53* alterations alone can justify tissue biopsy—even if liquid biopsy is feasible—to confirm the new histology [240]. Interestingly, inhibiting enhancer of zeste homologue 1/2 (encoded by *EZH1/2*) can curb the development of squamous features, resensitizing resistant squamous-like tumors to osimertinib [241]. Identifying early subclones that drive resistance before clinical progression is key for tailoring combination treatments [234]. However, no specific guidelines exist for managing EGFR-mutant NSCLC with histologic transformation, and standard chemotherapy regimens based on the new histology often yield limited benefits [239]. Emerging approaches, such as combining the PARP inhibitor niraparib with the anti–PD-L1 agent durvalumab (NCT04538378), are under investigation for SCLC-transformed EGFR-mutant NSCLC.

Epithelial-to-mesenchymal transition (EMT) is a key mechanism underlying drug resistance, as it enables epithelial cancer cells to acquire mesenchymal features that promote migration, invasion, and therapy evasion [242,243]. This shift is driven by EMT-inducing transcription factors (EMT-TFs)—including the ZEB, Snail, Slug, and Twist families—that downregulate epithelial markers (e.g., E-cadherin) and upregulate mesenchymal markers (e.g., vimentin) [244,245,246,247,248]. Furthermore, exosomal transfer of miR-210-3p and miR-17-5p from osimertinib-resistant cells to sensitive cells facilitates EMT, highlighting the importance of the tumor microenvironment (TME) in drug resistance [249,250]. These findings suggest that targeting EMT-TFs, which contribute to resistance when overexpressed, may offer promising therapeutic strategies [251].

Overexpression of ID1, a transcription factor, has also been implicated in osimertinib resistance via downregulation of E-cadherin and upregulation of vimentin [252]. Furthermore, mounting data indicate that standard treatments often fail to eliminate EMT-committed cells [209,210]. Targeting TGFβ signaling, which promotes EMT, has shown promise for overcoming osimertinib resistance, as TGFβ2 upregulation can activate SMAD2 and trigger mesenchymal features in resistant cells [243,253,254]. Inhibiting NF-κB—another pathway critical for resistant cell survival—also exerts cytotoxic effects [253]. Taken together, these findings underscore the multifaceted nature of treatment resistance—spanning histologic transformation, EMT, and intercellular communication—and highlight the need for innovative, multi-pronged strategies to improve outcomes for patients with EGFR-mutant NSCLC.

**Table 2 cancers-17-00459-t002:** Off-target mechanisms of osimertinib resistance and therapeutic strategies in EGFR-mutant NSCLC.

Mechanism Category		Approximate Prevalence	Therapeutic Strategies	References
**MET amplification**		Up to 8–19% of resistance cases; more frequent in first-line osimertinib	MET inhibitors (crizotinib, capmatinib) ± EGFR TKIs Combination therapy (e.g., afatinib + MET inhibitor)	[32,40,72,74,96,104,122,123,124,125,126,127,128,129,130,131,132,133,134,135,136,137,138,139,140,141,142,143,144,145,146,147,148,149,150]
**HER2 amplification and HER3 upregulation**	HER2 amplification/mutation	~2–5% in post-osimertinib setting (higher in second-line)	Anti-HER2 mAbs (e.g., trastuzumab) or ADCs (T-DM1) in combination with osimertinib Pan-HER inhibitors under evaluation	[32,40,119,120,151,152,153]
	HER3 upregulation	Frequently seen in resistant tumors; exact incidence unclear	HER3-targeted ADCs (e.g., patritumab deruxtecan) Combinations with EGFR TKIs	[128,154,155,156,157,158,159,160,161,162,163,164,165,166]
**Alterations in proliferation and apoptosis**	FGFR1 amplification	Rare; exact rates vary	GFR inhibitors ± osimertinib or AKT blockade Triple blockade (EGFR + AXL + FGFR) in preclinical models	[167,168,169,170]
	IGF1R activation	Rare; data mostly preclinical	IGF1R inhibitors in combination With osimertinib	[171,172,173,174]
	AXL overexpression	Frequently upregulated in resistant clones	AXL inhibitors (e.g., anlotinib) ± osimertinib Dual or triple targeting (AXL/MET/FGFR)	[15,168,170,175,176,177,178,179,180,181,182,183,184]
	BCL-2 family dysregulation	BIM deletion polymorphism in ~21% of East Asians	HDAC inhibitors to restore BIM Navitoclax (BCL-2/BCL-xL inhibitor) + osimertinib	[185,186,187,188,189]
	Cell cycle pathway alterations	Common in relapse; often correlated with shorter PFS	CDK4/6 inhibitors (abemaciclib, palbociclib) + osimertinib PLK1 inhibitors combined with EGFR TKIs	[190,191,192,193,194,195,196]
**Abnormal activation of downstream proliferative signaling pathways**	PI3K/AKT/mTOR-driven	Mutations in PIK3CA in ~2–5%; PTEN loss also reported	PI3K or mTOR inhibitors ± osimertinib Restoring or stabilizing the PTEN function	[92,197,198,199,200,201,202]
	RAS/RAF/MEK/ERK (MAPK)-driven	KRAS/NRAS/BRAF mutations in ~1–3% post-osimertinib	Targeted therapies (sotorasib, adagrasib for KRAS G12C) MEK inhibitors (trametinib, selumetinib) in combination with osimertinib	[40,50,96,104,113,168,204,205,206,207,208,209,210,211,212,213,214,215,216,217,218,219,220,221,222,223,224,225,226,227,228,229]
**Histologic transformation and epithelial-to-mesenchymal transition (EMT)**	Histologic transformation	2–15% of resistance cases vary with line of therapy	Biopsy confirmation needed Standard chemo- or immunotherapy based on new histology	[78,85,91,230,231,232,233,234,235,236,237,238,239,240,241]
	Epithelial-to-mesenchymal transition (EMT)	Common in resistant tumors, the exact incidence varies	TGFβ pathway blockade, NF-κB inhibition Targeting EMT transcription factors	[242,243,244,245,246,247,248,249,250,251,252,253,254]

ADC, antibody–drug conjugate; FGFR, fibroblast growth factor receptor; HDAC, histone deacetylase; IGF1R, insulin-like growth factor 1 receptor; mAb, monoclonal antibody; mTOR, mechanistic target of rapamycin; PI3K, phosphatidylinositol 3-kinase; PTEN, phosphatase and tensin homologue; T-DM1, trastuzumab emtansine.

## 5. Osimertinib-Based Combination Therapies: Exploring Synergistic Approaches

As aforementioned, osimertinib has shown remarkable efficacy as a third-generation EGFR-TKI in the treatment of EGFR-mutant NSCLC. Its integration into therapeutic regimens highlights its potential to enhance clinical outcomes when combined with other treatment modalities. However, the inevitable development of resistance to osimertinib remains a critical clinical challenge, necessitating innovative strategies to overcome this limitation. Consequently, various combination approaches (Table 3) have been investigated with the dual objectives of achieving additive or synergistic therapeutic effects and delaying the emergence of resistance, with the ultimate aim of optimizing patient outcomes in the context of EGFR-mutant NSCLC.

### 5.1. Chemotherapy

Preclinical work suggests that adding pemetrexed or cisplatin to osimertinib can help forestall drug resistance in NSCLC, though this approach may increase tissue fibrosis [255]. One mechanism involves ATP-binding cassette subfamily B member 1 (ABCB1, also known as P-glycoprotein), which promotes chemoresistance by expelling intracellular drugs. Notably, osimertinib has been shown in vitro to reverse ABCB1-mediated resistance to agents such as paclitaxel, colchicine, and vincristine, suggesting that EGFR-TKI–chemotherapy combinations could be beneficial [256]. Beyond standard chemotherapy, a T7 peptide–modified nanocarrier co-delivering osimertinib and doxorubicin demonstrated promising blood–brain barrier penetration in preclinical models, leading to enhanced efficacy against brain metastases [257]. While these findings point to potential advantages in delaying resistance and achieving prolonged disease control—especially in patients with CNS involvement—more phase III evidence is required to establish combination therapy as a standard treatment.

Following osimertinib’s approval as first-line therapy, efforts have intensified to incorporate it into combination regimens. Clinically, adding platinum–pemetrexed to osimertinib is safe and appears to control central nervous system (CNS) metastases [28], although initial studies did not show a significant PFS benefit over osimertinib monotherapy (15.8 vs. 14.6 months; *p* = 0.83) [258]. The phase III COMPEL trial (NCT04765059) is examining platinum–pemetrexed plus osimertinib in patients with non-CNS progression after an initial osimertinib response [259], and the phase II EPONA trial in Japan is focusing on this combination for individuals with EGFR-mutant, brain-metastatic NSCLC [260]. Both aim to determine whether such strategies can further extend PFS and OS. Early-phase results are encouraging: the phase II OPAL trial reported an ORR of 90.9% and a median PFS of 31.0 months for first-line osimertinib combined with platinum–pemetrexed [261]. FLAURA2 (NCT04035486), a larger phase III study, has provided preliminary safety data indicating no unexpected toxicities among 30 participants treated with the combination—mirroring the tolerable safety profile seen in OPAL [41]. Its final phase III data demonstrated a significant PFS benefit (HR 0.62; 95% CI 0.49–0.79; *p* < 0.001) compared with osimertinib monotherapy, alongside a higher 24-month progression-free rate (57% vs. 41%). The ORR increased slightly (80% vs. 76%), though adverse events rose as well, largely attributable to chemotherapy [41,262].

An alternative approach explored by the JCOG1404/WJOG8214L trial (UMIN000020242) involved intermittent platinum-doublet chemotherapy after initial EGFR-TKI therapy [263]. Patients were randomized to continuous EGFR-TKI monotherapy or an intermittent regimen of gefitinib or osimertinib followed by cisplatin–pemetrexed. Although median OS was similar in both arms (48.0 months), PFS was superior with intermittent chemotherapy (HR 0.762; *p* = 0.0003), suggesting that delaying resistance may be feasible, even without a survival advantage [263]. ctDNA analysis has emerged as an important predictive tool in EGFR-mutant NSCLC. In retrospective assessments of AURA3 and FLAURA, patients lacking detectable baseline EGFR mutations in ctDNA had notably prolonged PFS (HR 0.48 and 0.54, respectively), and those who achieved early plasma EGFR mutation clearance by week 3 of treatment showed extended PFS [37].

### 5.2. VEGF Inhibitor-Based Therapy

Combining EGFR TKIs with VEGF pathway inhibitors is promising for EGFR-mutant NSCLC, as EGFR activation promotes VEGF expression via overlapping signaling pathways [264]. Early trials using erlotinib plus bevacizumab (e.g., JO25567, NEJ026, ARTemis/CTONG1509) demonstrated significantly improved progression-free survival, although overall survival gains were less consistent [265,266,267]. Extending this strategy to third-generation TKIs, particularly osimertinib, yielded encouraging, yet variable results. Early-phase research indicated that osimertinib plus bevacizumab could delay resistance, with a phase I/II study reporting a 12-month PFS rate of 76% and an 80% ORR [268,269]. Nonetheless, larger randomized trials provided conflicting findings. The Japanese phase II WJOG9717L trial did not detect a significant PFS benefit (22.1 vs. 20.2 months; HR, 0.862) for osimertinib plus bevacizumab in patients with sensitizing EGFR mutations [270]. Similarly, the European phase II BOOSTER trial found no PFS advantage (15.4 vs. 12.3 months; HR 0.96) for T790M-positive patients, with higher rates of grade ≥ 3 adverse events such as hypertension and proteinuria [271]. Despite these mixed outcomes, the phase II FLAIR study is ongoing, evaluating osimertinib plus bevacizumab in patients harboring L858R mutations, with results anticipated in 2025 [272].

By contrast, combining EGFR TKIs with ramucirumab, a VEGF receptor-2-targeting monoclonal antibody, has been more promising. The phase III RELAY trial demonstrated significantly improved PFS when erlotinib was combined with ramucirumab (19.4 vs. 12.4 months; HR, 0.59; *p* < 0.0001) [273]. In the phase II RAMOSE trial, osimertinib plus ramucirumab likewise showed a notable PFS extension over osimertinib monotherapy (24.8 vs. 15.6 months; HR, 0.55), with a 12-month PFS rate of 76.7% versus 61.9% [274]. Toxicities remained manageable, and the discrepancy in efficacy between bevacizumab and ramucirumab combinations may reflect differences in drug mechanisms and toxicity profiles [274].

Overcoming resistance to osimertinib remains a key objective, and targeting the tumor vasculature in the tumor microenvironment (TME) with anti-angiogenic agents (e.g., anlotinib, a multi-target TKI that targets VEGFR, FGFR, PDGFR, and c-kit) offers a potential solution [275]. A retrospective analysis of 111 patients found that adding anti-angiogenic therapy to osimertinib significantly improved PFS (9.84 months) and OS (16.79 months) compared with chemotherapy alone or chemotherapy plus immune checkpoint inhibitors [276]. Preclinical models showed augmented infiltration of CD4+ and PD-1+/CD8+ T cells and reduced tumor-associated macrophages secreting IL-1β and CCL18 when VEGFR inhibition was introduced [276,277]. Single-cell RNA sequencing (scRNA-seq) further corroborated these immunomodulatory effects, identifying enhanced cytotoxic T-cell populations and fewer TAM-associated markers [276]. These results highlight the role of anti-angiogenesis therapies in promoting the antitumor immune response. Clinically, high VEGFR2 or VEGF-C expression has been correlated with diminished osimertinib sensitivity, underscoring the rationale for VEGF pathway blockade to improve outcomes [278].

### 5.3. Radiotherapy

Radiotherapy (RT) was extensively employed alongside first-generation EGFR TKIs (e.g., gefitinib, erlotinib) [279], but its role with third-generation EGFR TKIs, particularly osimertinib, is still evolving. Although roughly 41.25% of EGFR-mutant NSCLC recurs at the primary site and these tumors have a heightened propensity for brain metastases [280], many questions remain regarding optimal RT timing, dosing, techniques, and toxicity when combined with osimertinib [280,281]. Early clinical data suggest concurrent thoracic RT and osimertinib could increase rates of radiation pneumonitis, pointing to the need for cautious dose management [281,282]. Preclinical findings indicate that osimertinib boosts radiosensitivity, reinforcing the rationale for its use in combination with RT, provided proper dose guidelines are established [283].

Brain metastases pose a serious threat in EGFR-mutant NSCLC, with CNS-related mortality rates exceeding those observed in EGFR wild-type NSCLC (44.8% vs. 8.3%) [284]. While earlier-generation TKIs exhibit limited blood–brain barrier (BBB) penetration, RT or chemotherapy may transiently disrupt the BBB to enhance drug delivery. In contrast, osimertinib already has superior BBB permeability and robust intracranial efficacy [284], making its combination with cranial RT an appealing strategy to manage CNS progression. Retrospective analyses provide support: one study demonstrated improved OS for patients receiving osimertinib plus cranial RT (53 vs. 40 months, *p* = 0.014) [64]. Another investigation suggested that delaying RT until needed does not compromise outcomes (no significant differences in PFS or OS) and may reduce radiation-induced side effects [285].

Multiple trials are exploring the benefit of consolidative or local RT with osimertinib. In the single-arm NCT04764214 study, patients with metastatic EGFR-mutant NSCLC received stereotactic radiotherapy (SRT) following an initial response to osimertinib, achieving a median PFS of 29.9 months and manageable toxicity [286]. The phase II NORTHSTAR trial investigated local consolidative therapy—including RT or surgery—after osimertinib induction, finding no notable increase in serious adverse events [287]. Similarly, the NCT03595644 study reported that patients with residual oligometastatic disease who underwent stereotactic body radiation therapy (SBRT) after maximal tumor response to osimertinib experienced no disease progression over 18 months, with only mild toxicities [288]. Another phase II trial (NCT03667820) combining SBRT and continuous osimertinib reported a median PFS of 32.6 months and an OS of 45.7 months, surpassing historical outcomes for osimertinib monotherapy—which commonly deteriorates before two years [289].

### 5.4. MET-Inhibitor Based Therapy

As aforementioned, MET dysregulation is a major driver of acquired resistance to EGFR TKIs in NSCLC, prompting the development of diverse MET inhibitors. These agents include small-molecule inhibitors (e.g., savolitinib, crizotinib, cabozantinib, capmatinib, and tepotinib), MET receptor monoclonal antibodies (e.g., onartuzumab), and antibodies against the MET ligand HGF (e.g., ficlatuzumab and rilotumumab). Preclinical data suggest that combining osimertinib with crizotinib or capmatinib can overcome MET-mediated osimertinib resistance [290,291]. Additionally, capmatinib has been shown to inhibit MET/AKT/Snail signaling and reduce cancer-associated fibroblast (CAF) production, resensitizing lung cancer cell lines to osimertinib [150]. Case reports support these findings: for example, a patient harboring a MET Y1003N mutation achieved a partial response to combined capmatinib and osimertinib [291]. However, KRAS G12C subclones—detected in both xenograft models and patient samples—may emerge as a new mechanism of resistance when targeting EGFR or MET [108,249]. Other agents, including HQP8361, dictamnine, and the natural compound berberine, have likewise shown promise in countering MET-driven osimertinib resistance by inhibiting MET signaling and inducing tumor cell apoptosis [292,293,294].

Amivantamab (JNJ-61186372), a bispecific EGFR/MET antibody, is another important therapeutic option. Approved by the FDA for patients with *EGFR* exon 20 insertions [295], amivantamab also offers activity in tumors partially responsive to osimertinib [296]. In the CHRYSALIS study, it achieved a confirmed ORR of 40% and a median PFS of 8.3 months [297]. Sustainable responses have been reported in triple-mutant EGFR (L858R/T790M/G796S) tumors [298], and combining amivantamab with lazertinib has demonstrated synergistic efficacy [299,300,301]. Nevertheless, residual Src-family kinase activity can induce amivantamab resistance, highlighting the complexity of overcoming advanced disease [302]. The phase III MARIPOSA-2 study strengthens the case for amivantamab: in EGFR-mutated advanced NSCLC after progression on osimertinib, both amivantamab monotherapy and amivantamab–lazertinib significantly prolonged PFS compared to chemotherapy (6.3 and 8.3 vs. 4.2 months; *p* < 0.001) [303]. Another innovative approach, SHR-A1403—an antibody–drug conjugate linking an MET antibody to a microtubule-disrupting agent—has also shown potential against MET-overexpressing, osimertinib-resistant tumors [304].

Another key trial investigating these dual-target approaches is the multi-arm phase Ib TATTON study, which is evaluating osimertinib in combination with savolitinib, selumetinib (MEK1/2 inhibitor), and durvalumab (anti-PD-L1 antibody) in T790M-positive EGFR-mutant NSCLC [305,306]. Based on early results, a recommended regimen of osimertinib plus savolitinib demonstrated favorable safety and clinical outcomes for patients with MET-driven resistance [306]. Ongoing investigations include a phase II trial evaluating osimertinib plus savolitinib (NCT03778229) [307] and another testing osimertinib with tepotinib (NCT03940703) [308]. Canadian expert panels have further advocated MET inhibitors for MET-amplified, EGFR-mutant NSCLC, regardless of the treatment line [309]. Looking ahead, first-line strategies combining third-generation TKIs with MET inhibitors may offer even greater benefits in EGFR-mutant, MET-amplified NSCLC [299]. Anticipated results from several ongoing phase III trials—examining amivantamab with lazertinib or osimertinib—are expected to solidify dual EGFR/MET blockade as a standard option for advanced disease [310,311].

### 5.5. Immunotherapy

Combining PD-1/PD-L1 inhibitors (e.g., pembrolizumab, durvalumab, nivolumab) with osimertinib has drawn considerable attention as a strategy to overcome acquired resistance in EGFR-mutant NSCLC [312,313]. While immune checkpoint inhibitors (ICIs) have transformed cancer care, their efficacy in EGFR-mutant NSCLC progressing on EGFR TKIs is generally limited [314]. Consequently, there is growing interest in using ICIs after resistance to EGFR TKIs emerges [315,316]. One rationale is that osimertinib can augment CD8+ T-cell infiltration, potentially amplifying antitumor immunity [317,318]. However, trials such as TATTON showed an elevated risk of severe immune-related adverse events (irAEs), including interstitial lung disease (ILD), leading to early termination [229]. Similarly, the CAURAL trial was halted due to ILD [319]. Other notable toxicities—such as severe hepatotoxicity, colitis, and Stevens–Johnson syndrome—have been observed in patients receiving combined osimertinib–ICI therapies, a phenomenon less frequently reported with other EGFR TKIs plus immunotherapy [320,321].

Recent mechanistic studies offer insights into these outcomes. Elevated NADPH oxidase 4 (NOX4) appears to drive tumorigenesis and EGFR-TKI resistance via an IL-8–PD-L1 axis. Silencing NOX4 can resensitize resistant cells to both gefitinib and osimertinib [322]. Additionally, NOX4 overexpression increases YY1, which upregulates IL-8 and PD-L1, fostering immune escape [322]. Clinically, high NOX4 and IL-8 correlate with poorer responses to anti-PD-L1 therapy, while dual inhibition of NOX4 (with GKT137831) and EGFR has shown enhanced tumor suppression in mouse xenograft models [322]. Another study implicates macrophages in toxicity through IL-6/JAK/STAT3 signaling: although osimertinib reduces EGFR phosphorylation in macrophages, it paradoxically increases total EGFR and activates proinflammatory cytokine release [323]. Combining ruxolitinib, a JAK inhibitor, with osimertinib mitigates this inflammatory response and liver injury, highlighting a potential strategy to lower irAEs [323].

Timing also seems crucial. Patients receiving an ICI within three months prior to osimertinib faced a 24% risk of severe irAEs, often appearing around three weeks into treatment [324]. By contrast, those switching from other EGFR TKIs before adding an ICI had fewer serious adverse events [324]. Tumor PD-L1 expression likewise influences treatment outcomes: patients with PD-L1 ≥ 50% experience shorter PFS (about 9.7 months vs. 26.5 months in those below 50%), and higher PD-L1 expression has been linked to worse PFS and OS [325,326,327,328,329,330]. Case studies illustrate both efficacy and toxicity. One patient with *EGFR* exon 19 deletion/T790M achieved six months of PFS on osimertinib plus pembrolizumab [331], whereas others developed Stevens–Johnson syndrome shortly after starting osimertinib post-ICI, although a long hiatus allowed for eventual retreatment [321].

Beyond single-agent immunotherapy, combining PD-1 inhibitors with platinum-based chemotherapy has demonstrated durable responses in patients harboring T790M–cis-C797S double mutations [332], suggesting a role for immunotherapy in specific genomic settings. Hyperprogressive disease remains a concern in up to 21% of patients receiving ICIs [333,334]. Novel approaches include the use of IL-12, which reduced immunosuppression and restored osimertinib sensitivity in resistant preclinical models [335]. Moreover, certain “inflamed” EGFR-mutant tumors—characterized by dense CD8+ T-cell infiltration and few FOXP3+ Treg cells—may respond profoundly to ICIs, indicating that immune profiling at the point of resistance could guide therapy selection [336].

## 6. Conclusions

Osimertinib has substantially improved outcomes for patients with EGFR-mutant NSCLC across first-line, subsequent-line, and adjuvant settings. However, acquired resistance—driven by tertiary EGFR mutations, activation of bypass pathways (e.g., MET and HER2/HER3), and histologic transformation—continues to present a major challenge. Growing evidence suggests that combination strategies incorporating chemotherapy, VEGFR inhibitors, radiotherapy, MET inhibitors, and immunotherapies may address these resistance mechanisms, but they also carry increased risks of toxicity and necessitate robust biomarkers for appropriate patient selection. Future success in mitigating resistance will hinge on real-time molecular profiling techniques, such as liquid biopsy and single-cell analyses, to detect emerging resistant clones before clinical progression. Ongoing clinical trials of novel osimertinib-based regimens, though not yet formally published, are anticipated to offer critical guidance on refining these approaches. Ultimately, a personalized treatment paradigm that aligns regimens with each tumor’s evolving molecular landscape—and expands clinical trial designs to encompass real-world diversity—could not only delay or overcome resistance but also improve survival and quality of life for patients with EGFR-mutant NSCLC.

## Figures and Tables

**Figure 1 cancers-17-00459-f001:**
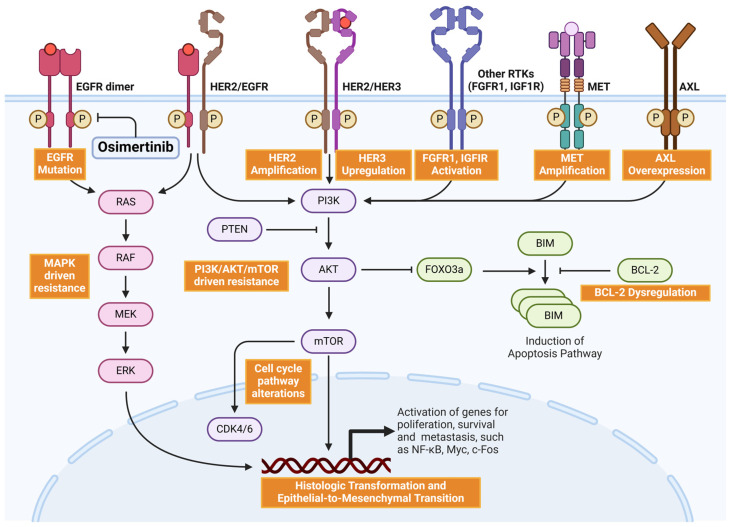
Key mechanisms of off-target resistance to osimertinib in NSCLC. Resistance arises from both on-target mechanisms involving EGFR and off-target pathways. Off-target resistance is driven by alternative signaling pathways such as MAPK (RAS-RAF-MEK-ERK) and PI3K/AKT/mTOR, activated by EGFR mutations, HER2 amplification, HER3 upregulation, MET amplification, AXL overexpression, abnormalities in other receptor tyrosine kinases (RTKs, such as FGFR1, IGF1R), BCL-2 dysregulation, cell cycle alterations (CDK4/6), histologic transformation, and epithelial-to-mesenchymal transition (EMT). Together, these mechanisms promote tumor survival, proliferation, and metastasis and reduce the efficacy of osimertinib. This figure was created using BioRender (https://BioRender.com, accessed on 20 January 2025) under a paid license.

**Table 3 cancers-17-00459-t003:** Ongoing osimertinib-based combination clinical trials in NSCLC.

Combination		Phase	Status	Clinical Trials Number
Radiotherapy	Stereotactic body radiation therapy (SBRT)	II	Active, not recruiting	NCT03667820
	SBRT	N/A	Not yet recruiting	NCT05583409
	SBRT	N/A	Recruiting	NCT05033691
	SBRT	N/A	Not yet recruiting	NCT05998993
Chemotherapy	carboplatin and pemetrexed	III	Recruiting	NCT04695925
	carboplatin and pemetrexed	II	Recruiting	NCT04410796
VEGF inhibitor	Bevacizumab	III	Recruiting	NCT04181060
	Bevacizumab	II	Not yet recruiting	NCT04988607
	Ramucirumab	III	Active, not recruiting	NCT02411448
MET inhibitor	Tepotinib	II	Active, not recruiting	NCT03940703
	Capmatinib, Nazartinib, and Gefitinib	II	Recruiting	NCT03040973
	Savolitinib	III	Recruiting	NCT05261399
	Savolitinib	III	Recruiting	NCT05015608
	Savolitinib	II	Active, not recruiting	NCT03778229
	Savolitinib	II	Active, not recruiting	NCT05163249
	Savolitinib	II	Active, not recruiting	NCT04606771
Others	Abemaciclib (CDK4/6 inhibitor)	II	Active, not recruiting	NCT04545710
	Itacitinib (JAK1 inhibitor)	I/II	Active, not recruiting	NCT02917993
	Selumetinib (MEK inhibitor)	II	Active, not recruiting	NCT03392246
	Sapanisertib (mTOR inhibitor)	I	Active, not recruiting	NCT02503722
Multi-drugs	AZD6094, Selumetinib	I	Active, not recruiting	NCT02143466
	AZD4547, Vistusertib, Palbociclib, Crizotinib, Selumetinib, Docetaxel, AZD5363, Durvalumab, Sitravatinib, AZD6738	II	Active, not recruiting	NCT02664935
	Savolitinib, Gefitinib, Necitumumab, Durvalumab, Carboplatin, Pemetrexed, Alectinib, Selpercatinib, Selumetinib, Etoposide, Cisplatin, Datopotamab deruxtecan	II	Active, not recruiting	NCT03944772

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
