# Peer review of "Optimizing Osimertinib for NSCLC: Targeting Resistance and Exploring Combination Therapeutics"

_cancers, 2025, doi:10.3390/cancers17030459_

Round 1
Reviewer 1 Report
Comments and Suggestions for Authors
Simple Summary is missing.
The background part can be more focused on EGFR and osimertinib and removed those well known old knowledge. This also applies for the abstract.
The main problem of this article is that the lack of appropriate subheadings when the authors are presenting a large amount of data. The authors should separate each resistance mechanism further into
a. Science behind the resistance mechanism
b. The epidemiology of the resistance mechanism
c. The treatment to target the resistance mechanism
It is highly recommended to shorten this very lenghty article and to present some of the findings in Tables or Figures. It is very tiring to read a 30 page review article, which is almost a book chapter. I will advise the authors to make the article much more concise.
Comments on the Quality of English LanguageNil
Author Response
Reviewer 1
Simple Summary is missing.
The background part can be more focused on EGFR and osimertinib and removed those well known old knowledge. This also applies for the abstract.
The main problem of this article is that the lack of appropriate subheadings when the authors are presenting a large amount of data. The authors should separate each resistance mechanism further into
- Science behind the resistance mechanism
- The epidemiology of the resistance mechanism
- The treatment to target the resistance mechanism
It is highly recommended to shorten this very lengthy article and to present some of the findings in Tables or Figures. It is very tiring to read a 30 page review article, which is almost a book chapter. I will advise the authors to make the article much more concise.
Answer
We appreciate the time you have taken to review our work and your insightful comments. Below is our point-by-point response to your comments and the corresponding revisions made to the manuscript:
- We have now incorporated a concise Simple Summary at the beginning of the manuscript. This summary briefly highlights the key findings and clinical implications of our review, providing a reader-friendly overview for both specialist and non-specialist readers.
- We agree that conciseness in the background section is important. We have revised both the Abstract and Background sections to focus more on EGFR and osimertinib. Redundant or well-established knowledge has been reduced or removed. We have also shortened the main text by consolidating overlapping discussions and omitting less critical details, without compromising essential content. This streamlining ensures that readers can quickly grasp the current status and rationale of osimertinib in NSCLC without excessive repetition of foundational information.
- We agree with the reviewer that appropriate subheadings are important. Accordingly, we added several subheadings to the main text, including additional subheadings under the section titled "On-Target (EGFR-Dependent) Mechanisms and Corresponding Therapeutic Strategies." Indeed, under each resistance mechanism subheading, we first described the epidemiology, followed by the corresponding treatments targeting the mechanism, as suggested by the reviewer. To further address the reviewer’s suggestions, we have transferred some descriptions into new tables (Tables 1 and 2), which visually represent key data on the science behind the resistance mechanisms, their epidemiology, and the treatments targeting each mechanism. This approach not only resolves the issue of lacking subheadings but also addresses another of the reviewer’s suggestions—namely, to present some findings in tables or figures.
In summary, by adding additional subheadings, reorganizing and streamlining redundant paragraphs, and creating new tables and a figure, we have significantly reduced the overall length while providing a more structured and accessible review.
Reviewer 2 Report
Comments and Suggestions for Authors
Summary:
This paper provides a comprehensive review of the evolving role of third-generation EGFR-targeted therapy, osimertinib, in the treatment of NSCLC. It delves into the mechanisms of osimertinib resistance in NSCLC, including both on- and off-target resistance mechanisms. The paper also examines the potential benefits of combining osimertinib with radiotherapy, anti-angiogenic agents, immune checkpoint inhibitors, and other molecularly targeted therapies. By summarizing preclinical and clinical evidence, the review highlights how these combination strategies may enhance osimertinib's efficacy, prolong patient survival, and maintain favorable safety profiles.
This timely and relevant review addresses a critical gap in the treatment of EGFR-mutant NSCLC by offering insights into extending the therapeutic benefits of osimertinib through combination approaches.
Major comments:
1. The article should include a Simple Summary and keywords.
2. It is recommended to incorporate the LAURA study on Osimertinib after Chemoradiotherapy in Stage III EGFR-Mutated NSCLC before discussing the ADAURA trial in the background section of the introduction.
3. For the resistance mechanism of osimertinib, it is suggested to briefly introduce both on- and off-target resistance mechanisms.
4. The inclusion of figures or tables to summarize and illustrate key points, such as resistance mechanisms or the clinical studies of combination therapies, would greatly improve the clarity and readability of the paper.
5. Tables summarizing completed clinical studies and ongoing trials without results would provide a more comprehensive understanding of the clinical research progress in this field of combination therapies.
Minor comments:
6. The format of headings and subheadings should be unified for enhance clarity and readability.
7. Please carefully review the article for minor errors, such as an extra "1." on page 4, line 191.
8 The numbering of each digit in the reference section is repeated.
Author Response
Reviewer 2
This paper provides a comprehensive review of the evolving role of third-generation EGFR-targeted therapy, osimertinib, in the treatment of NSCLC. It delves into the mechanisms of osimertinib resistance in NSCLC, including both on- and off-target resistance mechanisms. The paper also examines the potential benefits of combining osimertinib with radiotherapy, anti-angiogenic agents, immune checkpoint inhibitors, and other molecularly targeted therapies. By summarizing preclinical and clinical evidence, the review highlights how these combination strategies may enhance osimertinib's efficacy, prolong patient survival, and maintain favorable safety profiles.
This timely and relevant review addresses a critical gap in the treatment of EGFR-mutant NSCLC by offering insights into extending the therapeutic benefits of osimertinib through combination approaches.
Major comments:
1. The article should include a Simple Summary and keywords.
- It is recommended to incorporate the LAURA study on Osimertinib after Chemoradiotherapy in Stage III EGFR-Mutated NSCLC before discussing the ADAURA trial in the background section of the introduction.
- For the resistance mechanism of osimertinib, it is suggested to briefly introduce both on- and off-target resistance mechanisms.
- The inclusion of figures or tables to summarize and illustrate key points, such as resistance mechanisms or the clinical studies of combination therapies, would greatly improve the clarity and readability of the paper.
- Tables summarizing completed clinical studies and ongoing trials without results would provide a more comprehensive understanding of the clinical research progress in this field of combination therapies.
Minor comments:
6. The format of headings and subheadings should be unified for enhance clarity and readability.
- Please carefully review the article for minor errors, such as an extra "1." on page 4, line 191.
8 The numbering of each digit in the reference section is repeated.
Answer
We appreciate your thorough review, which has helped us strengthen our paper. Below, please find our point-by-point responses to your comments and a summary of the revisions we have made.
Response for Major Comments:
- We have included a Simple Summary at the beginning of the manuscript (following the abstract) to provide a clear and concise overview of our paper’s main points. We have also included a set of keywords immediately after the Simple Summary to assist in indexing and searchability.
- We have revised the background section to discuss the LAURA study before the ADAURA trial (in page 2, line 71 to 74), highlighting the sequence of relevant clinical trials and providing a more logical flow of information regarding osimertinib in different treatment settings.
- We have revised the headings in the section describing osimertinib resistance mechanisms. The revised text now clearly distinguishes on-target (e.g., tertiary EGFR mutations) and off-target (e.g., MET amplification, HER2 amplification, and histological transformations) resistance mechanisms, providing a more complete overview of how resistance emerges.
- In response to this suggestion, we have added three tables (Tables 1-3) presenting the overview of resistance mechanisms and the corresponding therapeutic strategies, as well as the related ongoing clinical trials. In addition, we have provided a figure (Figure 1) to depict the signaling pathways involved in resistance to osimertinib. We believe these additions will help readers navigate the main concepts more easily.
- We agree that presenting completed and ongoing trials separately will enhance clarity. In the revised manuscript, we have introduced three tables: Table 1 and Table 2 summarize the completed trials examining the on- and off-target resistance mechanisms of osimertinib and their corresponding therapies, respectively. Table 3 summarizes the ongoing clinical trials (without results) evaluating novel combinations with osimertinib. These tables include key details such as trial phase and design, offering a more comprehensive understanding of the progress in clinical research on combination therapies.
Response for Minor Comments:
- We have unified the format of the headings and subheadings throughout the manuscript, ensuring a consistent style and structure.
- We have thoroughly proofread the revised version of the manuscript and removed the extra “1.” on page 4, line 191 (now on page 5, line 185 of the revised manuscript).
- We have corrected the repeated numbering issue in the reference section so that each reference is now numbered only once in the reference list.
Reviewer 3 Report
Comments and Suggestions for Authors
I think this is an excellent paper that covers overcoming osimertinib-resistance in EGFR-mutated lung cancer.
There is no simple summary.
There is no simple summary.
Line 733 shows Stevens-Johnson syndrome, but line 771 shows the abbreviation SJS for Stevens-Johnson syndrome. Do you want to say SJS in line 733?
Author Response
Reviewer 3
Comments and Suggestions for Authors
I think this is an excellent paper that covers overcoming osimertinib-resistance in EGFR-mutated lung cancer.
There is no simple summary.
There is no simple summary.
Line 733 shows Stevens-Johnson syndrome, but line 771 shows the abbreviation SJS for Stevens-Johnson syndrome. Do you want to say SJS in line 733?
Answer
Thank you for your valuable feedback and for recognizing the relevance of our work on overcoming osimertinib resistance in EGFR-mutated lung cancer. We appreciate your comments and have addressed each point as follows:
- We apologize for the oversight. We have now added a Simple Summary at the beginning of the revised manuscript to succinctly highlight our objectives, key findings, and the clinical relevance of our review.
- Regarding to our omission of adding the abbreviation “SJS” in parentheses following the full term “Stevens-Johnson syndrome” when it first appears in the text (in line 733 of the previous version of our manuscript), we decide to write out the full term in both instances for clarity, as “Stevens-Johnson syndrome” only appears twice in the text.
Reviewer 4 Report
Comments and Suggestions for Authors
The authors have prepared a review article on resistance in EGFR-positive lung cancer. Overall, it is well organized, but a few points need revision. First, please describe the resistance mechanisms of osimertinib, specifically regarding on-target and off-target mechanisms, and re-examine the references you have cited. In addition, to make your ideas more precise, please revise the article by incorporating tables and figures to help readers understand more easily. I do not consider a review article without any tables or figures valid, nor would I read it. A review article is not merely an assemblage of text.
Author Response
Reviewer 4
The authors have prepared a review article on resistance in EGFR-positive lung cancer. Overall, it is well organized, but a few points need revision. First, please describe the resistance mechanisms of osimertinib, specifically regarding on-target and off-target mechanisms, and re-examine the references you have cited. In addition, to make your ideas more precise, please revise the article by incorporating tables and figures to help readers understand more easily. I do not consider a review article without any tables or figures valid, nor would I read it. A review article is not merely an assemblage of text.
Answer
Thank you for your constructive feedback on our review article. We appreciate the time and effort you took to evaluate our work and provide insights that will help us strengthen our manuscript. Please find our point-by-point responses below:
- We have revised the section covering osimertinib resistance to distinguish more clearly between on-target mechanisms (e.g., secondary EGFR mutations like C797S) and off-target mechanisms (e.g., MET amplification, HER2 amplification, histological transformations). This update now provides a more in-depth explanation of how osimertinib resistance develops in EGFR-mutated lung cancer, supported by key references in the literature.
- We have re-examined our references to confirm their accuracy, relevance, and completeness. In the process, we corrected some errors and added newly cited clinical trials to ensure our article reflects the most recent findings on mechanisms of resistance and combination therapeutic strategies.
- We agree that incorporating tables and figures will greatly help readers understand the topic more easily. We have introduced one figure and three tables: Table 1 and Table 2 summarize completed studies examining the on- and off-target resistance mechanism of osimertinib, respectively. Table 3 summarizes ongoing clinical trials (without results) evaluating novel combinations with osimertinib. These tables compile the pertinent details (trial phase, design, etc.) and offer a cohesive view of the current research landscape.
Round 2
Reviewer 1 Report
Comments and Suggestions for Authors
The article is much easier to read by including tables and figures, as well as cutting down the number of words.
Reviewer 4 Report
Comments and Suggestions for Authors
The authors addressed my comments, and I believe their paper has been revised for the better.